# The Pierre Auger Observatory: Review of Latest Results and Perspectives

**Dariusz Góra** [1,*] **, for the Pierre Auger Collaboration** [2,†,‡]

1    Institute of Nuclear Physics Polish Academy of Sciences, Radzikowskiego 152, 31-342 Cracow, Poland
2    Observatorio Pierre Auger, Av. San Martín Norte 304, Malargue 5613, Argentina
*    Correspondence: Dariusz.Gora@ifj.edu.pl
†    auger_spokespersons@fnal.gov.
‡    Full author list: http://www.auger.org/archive/authors_2018_07.html.

**Abstract:** The Pierre Auger Observatory is the world's largest operating detection system for the observation of ultra high energy cosmic rays (UHECRs), with energies above $10^{17}$ eV. The detector allows detailed measurements of the energy spectrum, mass composition and arrival directions of primary cosmic rays in the energy range above $10^{17}$ eV. The data collected at the Auger Observatory over the last decade show the suppression of the cosmic ray flux at energies above $4 \times 10^{19}$ eV. However, it is still unclear if this suppression is caused by the energy limitation of their sources or by the Greisen–Zatsepin–Kuzmin (GZK) cut-off. In such a case, UHECRs would interact with the microwave background (CMB), so that particles traveling long intergalactic distances could not have energies greater than $5 \times 10^{19}$ eV. The other puzzle is the origin of UHECRs. Some clues can be drawn from studying the distribution of their arrival directions. The recently observed dipole anisotropy has an orientation that indicates an extragalactic origin of UHECRs. The Auger surface detector array is also sensitive to showers due to ultra high energy neutrinos of all flavors and photons, and recent neutrino and photon limits provided by the Auger Observatory can constrain models of the cosmogenic neutrino production and exotic scenarios of the UHECRs origin, such as the decays of super heavy, non-standard-model particles. In this paper, the recent results on measurements of the energy spectrum, mass composition and arrival directions of cosmic rays, as well as future prospects are presented.

**Keywords:** Pierre Auger Observatory; ultra high energy cosmic rays; extensive air showers

## 1. Introduction

The measurement of ultra high energy cosmic rays (UHECRs), above $10^{17}$ eV, is a unique tool to provide answers to some of the most important questions in astrophysics: what are the sources of high energy cosmic rays (CRs); where are they produced; what are the corresponding acceleration mechanisms; and what is their elemental composition? In fact, although the existence of UHECRs has been experimentally proven for at least 50 years [1], these questions are still open, because at these energies, inference is limited by statistics. The flux of UHECR of energy above $10^{20}$ eV is estimated to be 0.5–1 events per square kilometer per century per steradian. The detection of CRs is also important because UHECRs may allow us to probe particle physics at an energy scale beyond TeV energies [2], probing center of mass equivalent energies of 500 TeV. When CRs arrive at Earth, they collide with nuclei in the atmosphere at center-of-mass energies, which are orders of magnitude above the ones available in man-made particle accelerators. CRs also propagate through the interstellar and intergalactic media and are thus subject to interactions with electromagnetic fields

and cosmic matter, thereby possibly providing indirect information about phenomena that arise only at large distances.

Cosmic rays collide with a nucleus high in the atmosphere, and the following hadronic interaction produces several energetic particles or secondaries, which, in their part, collide with other air nuclei, adding new energetic particles and developing a cascade or shower. Such showers are commonly called extensive air showers (EAS). The cascade (mostly pions, 80%, and kaons) grows from the first primary hadronic interaction until the energy per pion falls to the level where pions are likely to decay before colliding. Charged pions decay, producing atmospheric neutrinos and high energy muons, but neutral pions almost instantly decay to pairs of gamma-ray photons, feeding the electromagnetic component of the shower, which carries most of the initial energy.

The EAS can be continuously detected on the ground by large arrays of surface detectors (SD), which sample their lateral distribution every certain distance (1.5 km for the Auger Observatory), and/or fluorescence detectors (FD), which reveal the ultraviolet radiation (the fluorescence yield of air between 320 and 400 nm) emitted by the excitation of nitrogen during the passage of electromagnetic particles of EAS in the atmosphere. For very high energies (more than $10^{17}$ eV) of the primary particle, enough fluorescent light is produced so that the shower can be recorded from a distance of many kilometers by an appropriate optical detector system. The amount of fluorescent light is correlated with energy dissipated by shower particles and hence provides primary energy, together with a measure of the statistical uncertainty of each single shower event. However, the determination of the primary energy of EAS using the fluorescence detection technique requires an estimation of the energy carried away by particles that do not deposit all their energy in the atmosphere, i.e., of the so-called invisible energy, $E_{inv}$. It was calculated for the first time by Linsley [3], and it amounts to about 10–20% of the total shower energy. The estimation of invisible energy is typically made using Monte Carlo simulations and thus depends on the assumed primary particle mass and on model predictions for neutrino and muon production. For a given primary mass, the values of $E_{inv}$ predicted by the different hadronic interaction models differ by up to 30–40%.

In this paper, after a brief description of the Auger Observatory, we report recent results from the Observatory, i.e., the measurement of the UHECR spectrum, the anisotropy of arrival directions and the composition. Finally, limits on the UHE neutrino flux and limits on the flux of UHE photons are presented.

## 2. The Pierre Auger Observatory

The Pierre Auger Observatory [4,5], operating since 2004, is located near Malargue in the province of Mendoza, Argentina. It combines SD to measure secondary particles at the ground level together with FD to measure the development of air showers in the atmosphere above the array. This hybrid detection technique combines the calorimetric measurement of the shower energy through fluorescent light with the high-statistics data of the surface array.

The SD array consists of 1600 water-Cherenkov detectors arranged in a triangular grid with 1.5-km spacing (SD-1500 array) covering an area of about 3000 km$^2$. In addition, 61 detectors are distributed over 23.5 km$^2$ on a 750-m grid (SD-750 or 'infill' array); see Figure 1, left. Each water-Cherenkov detector has three photomultipliers (PMTs) on the top, which sample the shower signal. The signal detected at each station is expressed in a common calibration unit, which is called the vertical equivalent muon (VEM) [6]. The SD array is able to collect EAS at any time with almost a 100% duty cycle.

The FD consists of four telescope buildings (eyes) overlooking the detector array, as shown in Figure 1, left. Each building houses six telescopes with a $30° \times 28.6°$ field of view. The fluorescence light is focused in each telescope onto a camera consisting of 440 PMTs through its Schmidt-optics with a spherical mirror of ∼13 m$^2$. The fluorescence detector probes the longitudinal development of EAS (Figure 1, right) by measuring the photon emission from atmospheric nitrogen, which is excited by shower charged particles. It operates during clear moonless nights for a duty cycle of 15%. Three additional telescopes pointing at higher elevations (HEAT) are located near one of the FD

sites (Coihueco) to detect lower energy showers. An array of radio antennas, the Auger Engineering Radio Array (AERA) [5,7], complements the data with the detection of the shower radiation in the 30–80-MHz region. Details of the design and status of the Auger Observatory can be found in [4,5].

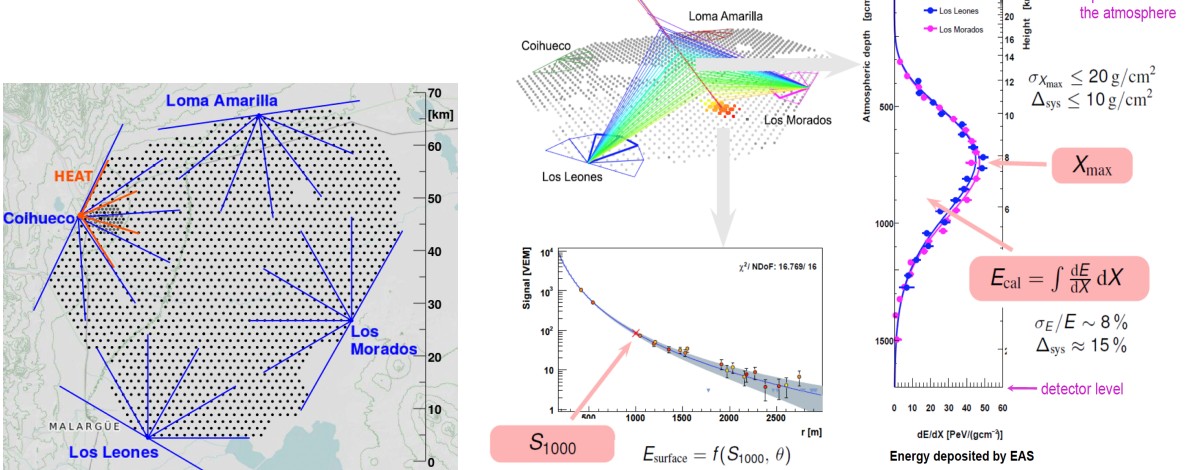

**Figure 1.** Left: Map of the Pierre Auger Observatory. The four telescope sites are marked in blue with lines indicating the field of view of the telescopes. The surface detector stations are represented by black dots. Right: Example of an event seen simultaneously by the surface detectors (SD) and fluorescence detectors (FD) (hybrid event). The right plot inset shows the measured energy deposit profile of an extensive air shower (EAS) from the FD, and the left plot shows the measured signal from stations of the SD with a fitted lateral distribution function (LDF). The LDF fit allows determining the particle density at a distance of 1000 m from the shower axis. VEM, vertical equivalent muon.

## 3. Results

### 3.1. High Energy Cosmic Ray Spectra

The hybrid nature of the Auger Observatory enables us to determine the energy spectrum of primary CRs without strong dependence on our limited knowledge of the mass composition and hadronic interaction models. The Auger approach is to use a selected sample of hybrid events (see Figure 1, right) in which the SD energy, $E_{SD}$, can be estimated using the FD. The calibration curves, which were used to find energies of SD events, are shown in Figure 2, left. The parameter chosen to characterize the size of an SD event for the SD-1500 array was the signal at 1000 m from the shower axis ($S(1000)$) [1], normalized to a mean zenith angle of the events of 38° ($S_{38}$) according to the method described in [10]. For the SD-750 array, the mean zenith angle of the events of 35° ($S_{35}$) was used, while for inclined showers (with the zenith angle larger than 60° seen by the SD-1500 array), the ($N_{19}$) energy estimator was used [11]. Following this method, the overall systematic uncertainty of the energy scale remained at 14% [12].

In Figure 2 (right), the Auger combined energy spectrum as presented at ICRC2017 is shown. The energy spectrum was obtained from the four spectrum components calculated for the hybrid events, events with the SD-1500/SD-750 array and inclined events. In total, 302,016 events were detected during the time of the measurements from January 2004–December 2016. The energy spectrum derived from hybrid data was combined with the one obtained from SD data using a maximum likelihood

---

[1]    The SD only samples the properties of an air shower at a limited number of points at different distances from the shower axis. To avoid the large fluctuations in the signal integrated over all distances caused by fluctuations in the shower development, Hillas [8] proposed to use the signal at a given distance $S(r)$ to classify the size of the shower. In [9], it has been shown that for the Auger array spacing (1.5 km), the optimum distance to minimize this experimental error is ~1000 m. Therefore, the observable that we use to relate to the primary energy will be the signal size at 1000 m e.g., $S(1000)$.

method [13]. Each event, according to the procedure explained in Figure 1, had its own response function, which had to be taken into account when doing the spectral fits. Following this method, all the spectrum components had the same energy scale. The overall systematic uncertainty of the energy scale came from the estimation of the fluorescence yield, the invisible energy, atmospheric conditions, reconstruction of the longitudinal profile of the showers and calibration, and it remained at 14%; see [12,14] for a detailed study of the systematics and statistical uncertainties of the Auger combined spectrum.

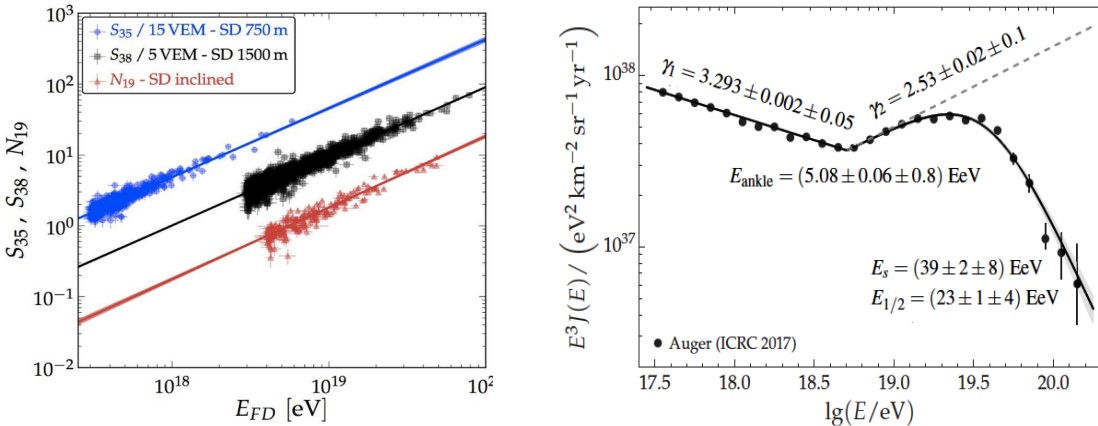

**Figure 2.** Left: Correlation between the SD energy estimators and the energy measured with the fluorescence telescopes ($E_{FD}$). The full line is the best fit to data. Right: The Auger energy spectrum with two empirical fit functions (see [12]). The fitted spectral indices and energies of the break and suppression are superimposed together with their statistical and systematic uncertainties.

In the plot, we illustrate two characteristic features of this combined spectrum: the so-called "ankle" and the flux suppression at the highest energies. The position of the ankle at $E_{ankle} = 5.08 \pm 0.06$(stat.) $\pm 0.8$(syst.) EeV had been determined by fitting the flux with a broken power law $\propto (E/E_{ankle})^{-\gamma}$ [12]. An index of $\gamma = 3.293 \pm 0.002$(stat.) $\pm 0.05$(syst.) was found below the ankle. A power law extension of the flux above the ankle was clearly excluded by the data, and we found a suppression energy of [2] $E_s = 39 \pm 2$(stat.) $\pm 8$(syst.) EeV with a spectral index $\gamma = 2.53 \pm 0.02$(stat.) $\pm 0.1$(syst.). The energy at which the integral flux dropped by a factor two below what would be expected without suppression was found to be $E_{1/2} = 23 \pm 1$(stat.) $\pm 4$(syst.) EeV. This value was considerably lower than $E_{1/2} = 53$ EeV, as predicted for the classical Greisen–Zatsepin–Kuzmin (GZK) scenario [15] in which the suppression at ultra high energies was caused by depletion of extra-galactic protons during propagation. UHECRs would interact with the CMB. The interaction would reduce their energy, so that particles traveling long intergalactic distances could not have energies greater than $5 \times 10^{19}$ eV. However, the suppression of the spectrum can also be described by assuming a mixed composition at the sources or by the limiting acceleration energy at the sources rather than by the GZK-effect [16,17].

### 3.2. Composition

The position of the maximum, $X_{max}$, was the atmospheric depth at which the number of particles in an extensive air showers reached its maximum. The $X_{max}$ depends on the primary nuclear mass; see as an example Figure 1, right. For a fixed energy, showers initiated by heavy nuclei reached their maximum size on average at smaller $X_{max}$ than those initiated by protons. Shower-to-shower

---

2    We fitted the flux with a power law allowing for a break in the spectral index at $E_{ankle}$ and a suppression of the flux at ultrahigh energies $\propto (1 + (E/E_s)^{\gamma})^{-1}$.

fluctuations in $X_{\max}$ were much less for heavy nuclei than for protons. Both of these properties were robust expectations that followed from a heavy nucleus being composed of many nucleons.

In Figure 3, we show the measured $\langle X_{\max} \rangle$ and the shower-to-shower fluctuations, $\sigma(X_{\max})$ from the Auger Observatory as also presented at ICRC 2017 [18]. The linear fit, $\langle X_{\max} \rangle = D_{10} \cdot \lg(E/eV) + c$, yielded an elongation rate (variation of $\langle X_{\max} \rangle$ per decade of energy) of $D_{10} = 79 \pm 1$ g/cm$^2$/decade between $10^{17.2}$ and $10^{18.33}$ eV; see Figure 3 (left). These values are larger than that of a constant mass composition of primary particle ($\sim$60 g/cm$^2$/decade), indicating that the mean primary mass became lower with increasing energy. At $10^{18.33 \pm 0.02}$ eV, the elongation rate became significantly smaller ($D_{10} = 26 \pm 2$ g/cm$^2$/decade), indicating that the composition became heavier with increasing energy. Furthermore the $\sigma(X_{\max})$ showed a similar behavior; see Figure 3 (right). As can be seen from the plot, the shower-to-shower fluctuations decreased from 60 g/cm$^2$ to about 30 g/cm$^2$ as the energy increased. These decreasing fluctuations were an independent signature of increasing average mass of the primary particles.

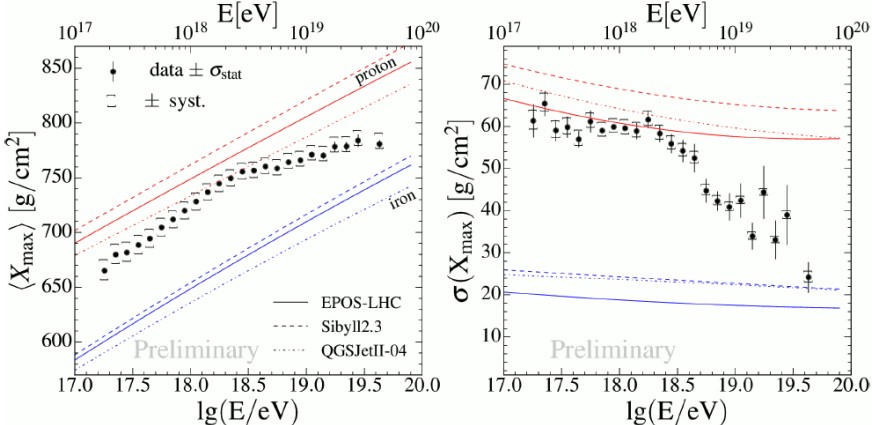

**Figure 3.** The mean $X_{\max}$ (left) and the standard deviation of the $X_{\max}$ (right) obtained by the FD as a function of the energy. The predictions of different hadronic interactions models, for pure proton and iron composition, are also shown (see [18]).

The Auger $X_{\max}$ data (distributions and their moments) enabled a step further in the interpretation of mass composition by studying the evolution of the first two moments of ln $A^3$ with energy [18,20,21]. The distributions of $X_{\max}$ in each energy bin can be compared to mixes of different primaries. It was found that the best agreement with the Auger $X_{\max}$ data could be achieved by fitting the mixes of proton (p), Helium (He), Nitrogen (N) and Iron (Fe) primaries [18,20–22] as obtained from CONEX [23] simulations using the post-LHC models EPOS-LHC [24], QGSJetII-04 [25] and Sybill 2.3 [26]. The different hadronic physics assumptions used in these models led to different $X_{\max}$ parametrizations. The model distributions were parametrized by a Gaussian convoluted with an exponential function, as described in [27]. The results of the fitted mass fractions are presented in Figure 4. It can be seen that Fe seemed to be almost absent over the whole energy range, except possible small fractions at the lowest and highest energies. There was a high percentage of protons for energies below $\sim 10^{19}$ eV, followed by an increase of the He fraction arising at $\sim 10^{18.6}$ eV and dropping at $\sim 10^{19.2}$ eV, succeeded by an increase of the nitrogen fraction at $\sim 10^{19.3}$ eV. However, the interpretation of $X_{\max}$ data depends on the hadronic interaction model. In particular, QGSJETII-04 appeared to be less consistent with data, as can be seen in the lower panel of Figure 4, where the $p$-value of the fit is shown.

---

3    This connection assumes the Heitler model of EAS [19].

This is because the present interpretation relies on air shower simulations that use hadronic interaction models to extrapolate particle interaction properties over two orders of magnitude in center-of-mass energy beyond the regime where they have been tested experimentally. A possible different interpretation can be that the proton-air interaction cross-section or multiplicity, or both, was increased, which would lead to a faster shower development, compared to heavy nuclei collisions [18]. In other words, there is still work to be done to interpret the Auger $X_{max}$ and $\sigma(X_{max})$ result.

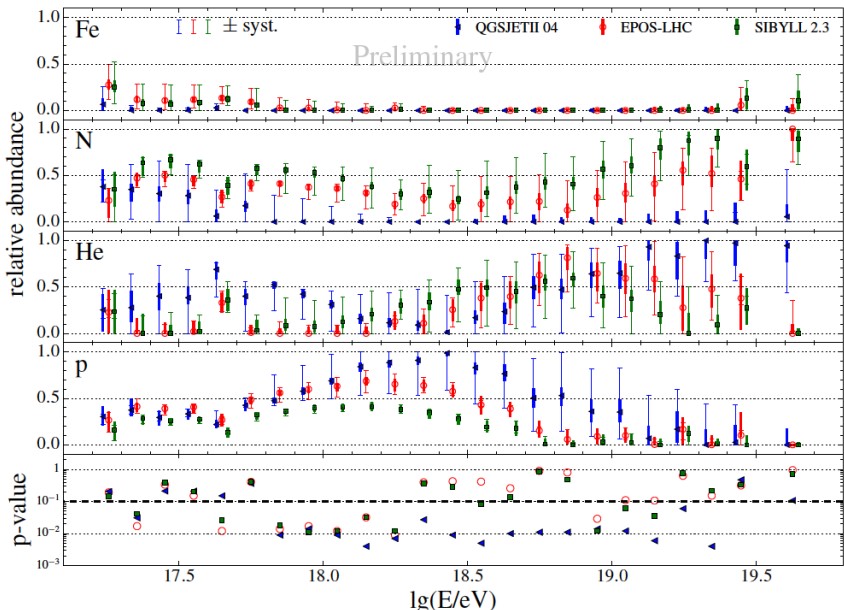

**Figure 4.** Mass fraction fits obtained using parameterizations of the $X_{max}$ distributions from the fluorescence $X_{max}$ data. The error bars indicate the statistics (smaller cap) and the systematic uncertainties (larger cap). The bottom panel indicates the goodness of the fits (*p*-values); see [18] for more details.

*3.3. Anisotropy*

An anisotropy in the arrival direction distribution of CRs in the energy range of the GZK suppression ($5 \times 10^{19}$ eV) was expected because of the highly anisotropic matter distribution due to galaxy clustering on distance scales of 100 Mpc. The extragalactic magnetic fields are not strong enough to deflect significantly high energy cosmic rays during their propagation to the Earth, so the observed source distribution should reflect the matter distribution in the nearby universe, to first order. The Auger Collaboration has undertaken several anisotropy searches at different energy ranges and angular scales. These use different tools, such as harmonic analysis, auto-correlation, correlation with source catalogs and the search for flux excesses in the visible sky and correlations with other experiments [28–30].

Among the most recent studies, the most exciting result was the observation of a large-scale anisotropy in the arrival directions of CRs above $8 \times 10^{18}$ eV [31]. Two energy bins, 4 EeV$< E \leq 8$ EeV and $E \geq 8$ EeV, were analyzed in the data obtained since the start of data taking with the Auger Observatory (total exposure of 76,800 km$^2$ sr y) to determine the amplitude of the first harmonic in right ascension; see Figure 5 (left). The events in the lower energy bin followed an arrival direction distribution consistent with isotropy, but in the higher energy bin, a significant anisotropy was found, with a *p*-value of $2.6 \times 10^{-8}$ under the isotropic null hypothesis, equivalent to a two-sided Gaussian significance of 5.6 $\sigma$. Penalizing for the fact that two energy bins were explored and for the past studies [28], where four additional lower-energy bins were examined, gave a significance to the observed dipole of 5.2 $\sigma$. The three-dimensional dipole, obtained by combining the first-harmonic

analysis in right ascension with a similar one in the azimuthal angle, had a direction in galactic coordinates (l, b) = (233°, −13°), which is about 125° away from the Galactic Center. Hence, this anisotropy indicates an extragalactic origin for these UHECR particles. The dipole anisotropy had an amplitude of $6.5^{+1.3}_{-0.9}\%$ and a level of significance of $5.2\,\sigma$. A sky map of the intensity of CRs arriving above 8 EeV is shown in Figure 5 (right).

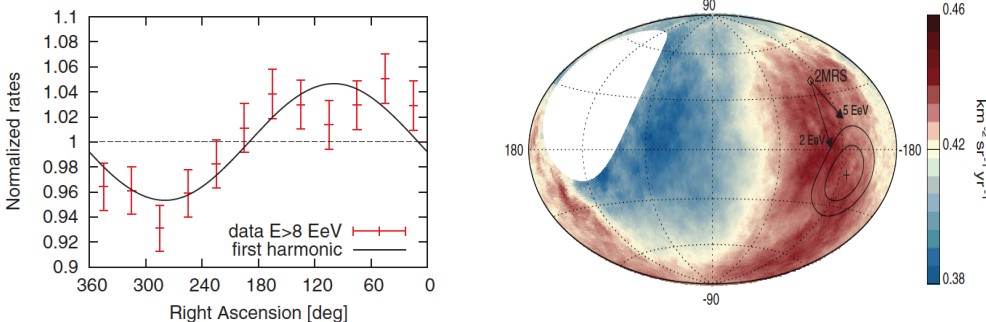

**Figure 5.** Left: Normalized rate of events as a function of right ascension for $E \geq 8$ EeV. The error bars represent $1\sigma$ uncertainties [31]. Right: Sky map in galactic coordinates of the cosmic ray flux for $E \geq 8$ EeV. The cross indicates the dipole direction, and the contours mark the 68% and 95% confidence level regions [31]. The diamond marks the dipole direction of the 2MRSgalaxy distribution [32]. The arrows show the deflections expected from the model described in [33] for particles with $E/Z = 5$ or 2 EeV.

### 3.4. Photons and Neutrinos

Another important issue concerning composition studies is the search for photons and neutrinos. Photons and neutrinos may directly trace back the origin of the UHECR. Essentially, all models of UHECR production predict neutrinos/photons as a result of the decay of charged/neutral pions produced in interactions of CRs within the sources themselves or while propagating through background radiation fields. For example, UHECR protons interacting with CMB give rise to "cosmogenic" or GZK neutrinos [34]. The cosmogenic neutrino flux is uncertain since it depends on the primary UHECR composition and on the nature and cosmological evolution of the sources, as well as on their spatial distribution; see Figure 6 (left). In general, about 1% of cosmogenic neutrinos are expected in the ultra high energy cosmic ray flux. However, the sensitivity to neutrinos is much lower, explaining no detection of neutrinos at UHE so far.

Due to their low interaction probability, neutrinos need to interact with a large amount of matter in order to be detected. One of the detection techniques is based on the detection of EAS in the atmosphere by looking for very inclined showers, which have an electromagnetic component leading to a broad time structure of the detected signal, in contrast to nuclei-induced showers, which for such inclinations, an electromagnetic component is attenuated.

When propagating through the Earth, only the so-called Earth skimming (ES) tau neutrinos may initiate detectable air showers above the ground [35,36]; in this case, tau neutrinos may interact within the Earth and produce charged tau leptons, which in turn decay into neutrinos with lower energies. Since the interaction length for the produced tau lepton is tens of kilometers at the energy of about 1 EeV, the leptons produced close to the Earth's surface may emerge from the ground, decay in air and produce EAS potentially detectable by the surface detector of the Auger Observatory [37]. The SD is also sensitive to down-going neutrinos (DG) in the EeV energy range. Down-going neutrinos of any flavor may interact through both charged (CC) and neutral current (NC) interactions, producing hadronic and/or electromagnetic showers (see [37] for details about the search technique). Up to now, no candidate events have been found by the Auger Observatory to fulfill the selection criteria. The absence of neutrino candidate events yielded an upper limit on the diffuse flux of neutrinos in the

EeV energy range [37], as shown in Figure 6 (left). It is also worth mentioning that predictions and upper limits from the different observatories shown in Figure 6 (left) are now on the same order of magnitude as the neutrino flux prediction for different models, which means that during the next few years, we can falsify these models. The present search allows putting strong constraints on several models predicting cosmogenic neutrinos in scenarios with proton models with strong source evolution and high redshift of the sources, at a 90% C.L.; see the red area in Figure 6 (left).

Photon-induced showers have a much lower muonic content, a smaller footprint on the ground and a deeper $X_{max}$, when compared to hadronic showers. The searches for photon candidates use methods that compare the characteristics of hadronic showers and photon-induced showers. Detailed procedures for photon searches using pure SD data can be found in [38].

The Auger Observatory has also set new photon limits with both the hybrid and SD detection methods [39,40]. The new limits are compared to previous results and to theoretical predictions in Figure 6 (right). In terms of the photon fraction, the current bound at 10 EeV approaches the percent level, while previous bounds were at the 10 percent level. The discovery of a substantial photon flux could have been interpreted as a signature of top-down (TD) models. The basic idea is that very massive (Grand Unified Theories scale) X particles decay, and the resulting fragmentation process downgrades the energy to generate the observed UHECR. Since the observed cosmic rays had energies orders of magnitude lower than the X particle mass, there were no problems with achieving the necessary energy scale. In turn, the experimental limits now put strong constraints on these models.

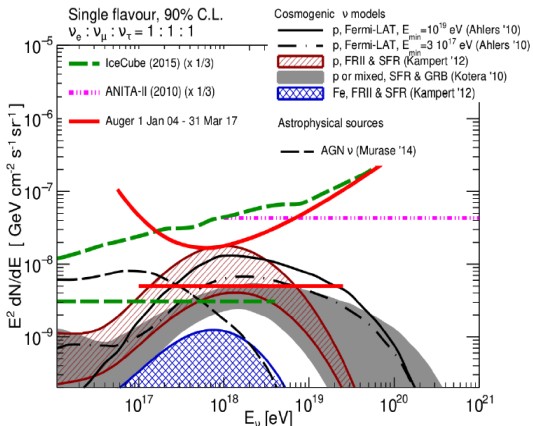 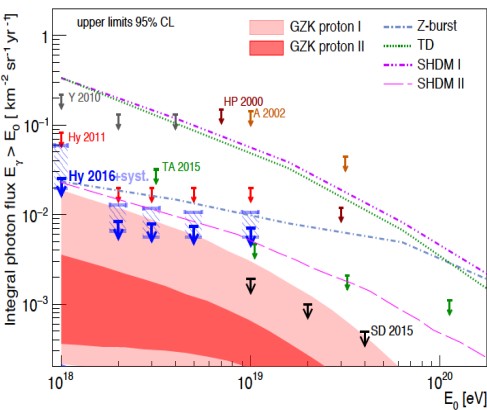

**Figure 6.** Left: Differential (curved red line) and integrated (straight red line) upper limits (90% C.L.) from the Pierre Auger Observatory for a diffuse flux of neutrinos in the period 1 January 2004–31 March 2017. The limits are drawn for a single flavor, assuming equal flavor ratios. Limits from ANITA(magenta dotted dashed line) and IceCube (green dashed line), along with several model predictions are also shown; see [37] for the complete set of references. Right: Upper limits on the integrated photon flux, along with several model predictions. The bold arrows (Hy2016) correspond to the most recent data analysis at Auger using hybrid events, and the blue dashed boxes mark the systematic uncertainties of this study. The limits of previous studies done by Auger, Hy2011 (red arrows) and SD 2015 (black arrows), the latter one using only SD data, are also shown. The results from other experiments are also presented. See [39,40] for the complete set of references. GZK, Greisen–Zatsepin–Kuzmin.

### 3.5. Multi-Messenger Astronomy

The Pierre Auger Observatory actively participates in multi-messenger searches in collaboration with other observatories. The detection of the first gravitational wave (GW) transient GW150914 on 14 September 2015, by the Advanced LIGO detectors, opened a new window into multi-messenger astronomy [41], enhancing the participation of many other experiments, of which the Pierre Auger Observatory is also a part. Auger performed neutrino searches in coincidence with the

gravitational wave events GW150914, GW151226, LVT15012 [42] and GW170817/GRB 170817A [43]. The GW170817/GRB 170817A [44], a gravitational wave event detected on 17 August 2017, was later observed as a short gamma-ray burst (GRB) by the Fermi-GBMand INTEGRAL. This event was caused by the merging of binary neutron stars in the host galaxy NGC4993 at a distance of 40 Mpc, the closest gravitational event detected so far. According to model predictions, from a typical GRB observed at different viewing angles, such a system may also accelerate cosmic rays to extreme energies, and thus emit photons and neutrinos up to GeV to EeV energies. The most promising neutrino-production mechanism from GRBs is related to the extended gamma emission, due to its relatively low Lorentz factor, resulting in high meson production efficiency; see [45] for more details. Neutrino searches related to this event were carried out by the three most sensitive neutrino observatories IceCube, Antares and Auger. The sky map of these neutrino searches is shown in Figure 7. In Auger, the whole $\pm 500$-s time window was observed in the ES channel field of view. No inclined showers passing the ES channel selection were detected during this period. Assuming neutrinos are emitted steadily during this period, with an energy spectrum of $E^{-2}$ [42], the non-detection of candidates allows us to put limits on neutrino fluence; see Figure 7 (right). In the following 14 days, searches were done both in the ES and DG channels. No neutrino candidates were found in this time window either. No significant counterpart was found in any of the searches with any of the observatories, a result which is compatible with the expectations of a GRB observed off-axis; see Figure 7 (right). Right now, predictions (Figure 7, right) for an NSmerger at 40 Mpc are about 1–4 orders of magnitude below the sensitivity for neutrinos. This implies that only events in the Milky Way or the Magellanic Clouds could be seen.

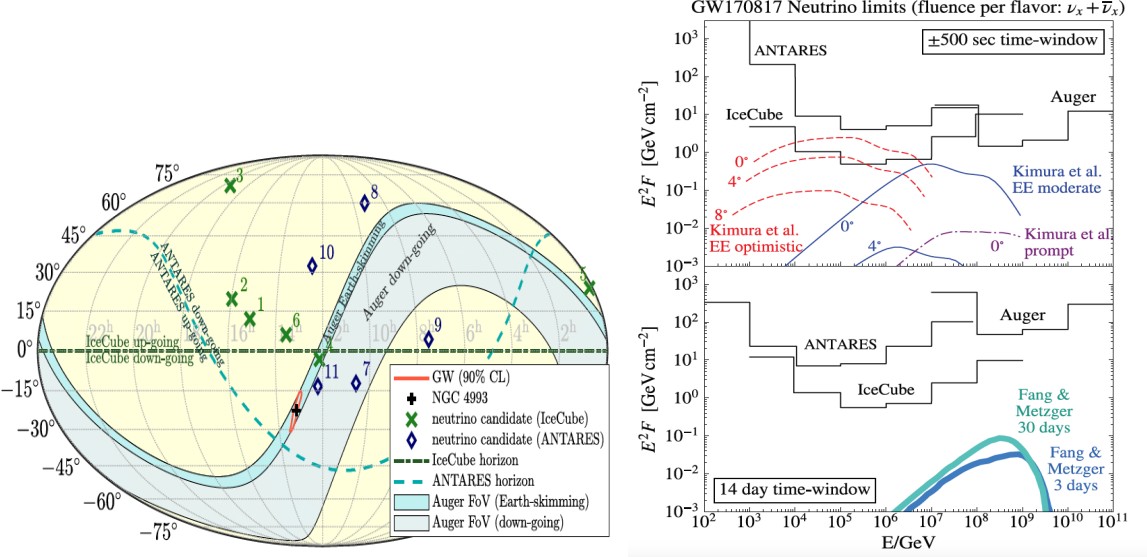

**Figure 7.** (Left) Sensitive sky areas of ANTARES, IceCube and Auger at the time of the GW170817 event in Equatorial Coordinates [42]. (Right) Upper limits at 90% C.L. of the neutrino spectral fluence from the GW1700817 event for a 500-s time window (top panel) and in the following 14 days after the trigger (bottom panel) [42].

### 3.6. Auger Prime

Taking stable data since 2004, the Pierre Auger Collaboration has published many results about the properties of UHECRs. Among the latest results at the highest energies using SD data, the extragalactic origin of cosmic rays with E > 8 EeV [31] was established. However, to identify the UHECR sources, it is crucial to determine the nuclear mass composition in the flux suppression energy region. The duty cycle of the FD of 15% allows collecting a significant data sample only for energies below $10^{19.6}$ eV [18]. Several other mass composition analyses using the SD were performed by the Pierre Auger Collaboration, but these suffer from larger systematic uncertainties due to the

uncertainties in the assessment of the muon content of the shower using the water-Cherenkov detectors. Thus, it is also difficult to build a consistent picture of the origin of UHECRs.

To address such challenges, the Pierre Auger Observatory is currently undergoing a major upgrade phase, called AugerPrime [46]. The key upgrade element is the installation of a 4-m$^2$ plastic scintillator detector on top of each of the 1600 water-Cherenkov stations (Figure 8), enabling a better discrimination between the electromagnetic and muonic components of the shower. Additionally, the duty cycle of the fluorescence telescopes will be extended, allowing a direct determination of the depth of shower maxima with increased statistics at the highest energies. The electronics of the SD stations is being upgraded to obtain an increased sampling rate and a better timing accuracy, as well as a higher dynamic range, allowing a better reconstruction of the geometry of the showers.

The first step for the ongoing upgrade is the AugerPrime Engineering Array (EA) [47], which consists of 12 upgraded detector stations already operational since 2016. With this setup, we have verified the basic functionality of the detector design, the linearity of the scintillator signal, the calibration procedures and operational stability. The construction of AugerPrime is expected to be finished in 2019.

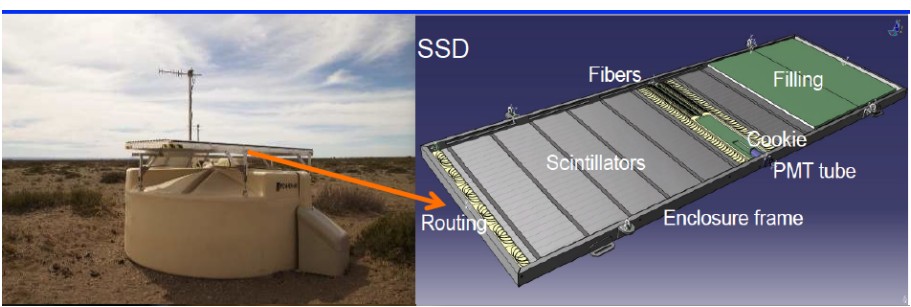

**Figure 8.** The picture of one upgraded station (left) and the layout of the surface scintillator detector (right). PMT, photomultiplier.

## 4. Conclusions

The measurements performed at the Pierre Auger Observatory indicate a change in the nature of cosmic rays at around 3 EeV and show a change in both the shape of the energy spectrum and of the elongation rate. These measurements add support to the hypothesis that an extragalactic component of mixed composition starts to dominate in this energy range. The recently observed dipole anisotropy at $E > 8$ EeV has an orientation that indicates an extragalactic origin of UHECRs. The near future particle accelerator results and AugerPrime will constrain the hadronic interaction models so that the interpretation of the evolution of the shower maximum with energy will be more conclusive. The photon limits exclude most of the top-down scenarios above 2 EeV. The neutrino limits, if no neutrino is observed, will improve by more than an order of magnitude and can constrain models of the cosmogenic neutrino production. These determinations, together with the arrival directions and mass composition analysis, will help in solving the puzzle of the origin of the highest energy cosmic rays.

**Funding:** This research was funded by the National Science Centre Grant No. 2016/23/B/ST9/01635, in Poland.

**Acknowledgments:** The successful installation, commissioning and operation of the Pierre Auger Observatory would not have been possible without the strong commitment and effort from the technical and administrative staff at Malargue and the financial support from a number of funding agencies in the participating countries, listed at https://www.auger.org/index.php/about-us/funding-agencies.

**Conflicts of Interest:** The authors declare no conflict of interest.

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
