# Peer review of "The Pierre Auger Observatory: Review of Latest Results and Perspectives"

_universe, doi:10.3390/universe4110128_

Round 1

Reviewer 1 Report

The paper consists of a review of the results and perspectives of the Pierre Auger Observatory, the world's largest detector of ultra-high energy cosmic rays. The author starts with a brief illustration of the main features that characterize the different detection systems of the experiment, which serves as an introduction to the discussion of the most important measurements that the Pierre Auger Observatory has performed so far. In addition to this, the author provides also an outlook on the future of the Observatory, more specifically on the AugerPrime upgrade.

I find the manuscript interesting and well written and I'm happy to recommend it for publication in Universe. Here below I just listen some typos that should be addressed before publication:

- line 3 : of their energy spectrum -> of the energy spectrum

- line 32 : gammas -> I would rather use "photons" or "gamma-rays"

- lines 42-43 : and on the composition -> and the composition

- line 43 : also limits -> limits

- line 88 : at highest energies -> at the highest energies

- line 92 : eV -> EeV

- line 150 and in the caption of Fig. 5 : galactic -> Galactic

Author Response

Dear Reviewer,

Thanks for commentes, their are included in the new version of the paper.

Best wishes
D. Gora

 -line 3 : of their energy spectrum -> of the energy spectrum

Text replaced in new version

- line 32 : gammas -> I would rather use "photons" or "gamma-rays"

Text replaced in new version

- lines 42-43 : and on the composition -> and the composition

Text replaced in new version

- line 43 : also limits -> limits

Text replaced in new version

- line 88 : at highest energies -> at the highest energies

Text replaced in new version

- line 92 : eV -> EeV

Text replaced in new version

- line 150 and in the caption of Fig. 5 : galactic -> Galactic

Text replaced in new version

Reviewer 2 Report

The draft of the article “The Pierre Auger Observatory: review of latest results and perspectives” provides a useful summary of the most recent measurements of ultra-high energy cosmic rays (UHERCs). The overall structure is adequate but the draft lacks precise connections to theoretical expectations, such as possible sources and acceleration mechanisms, and also general explanations of different terms in some sections. The text is understandable for expert readers but probably not for non-cosmic-ray-physicists.

Several sentences are not concise but use relativised words which are only understandable as part of the community. For such a review, precise statements should be used, and general conception should be worked out more.

The referee report is divided into two parts, with “major issues”, subdivided into the different sections of the paper, and “minor issues”, which may apply to the entire paper. I try to explain my issued points and mark the desired tasks in bold. Anything not bold are suggestions to make the paper more readable, also for not-specialists. The report contains the following points:

Major issues:

Abstract:

-          L6: “caused by the propagation of cosmic rays”: for those in the know, it is clear that the GZK cut-off is meant, however a few more words on this already in the abstract would help to understand the issue

-          L7: What is generally accepted as UHECR, i.e. at what energies?

-          L12: “super heavy particles” -> “super heavy, non-standard-model, particles”

Section 1:

-          L17/L23: “detection” -> “measurement”

-          L17: “… is important as it may allow us to answer …”; importance is relative to the reader, I would suggest: “… is a unique tool to provide answers to some of …”

-          L21: “for at last 50 years”; reference required.

-          L21: “mainly due to … in their understanding”; the sentence sounds not assuring, maybe: “because at these energies, inference is limited by statistics, (number here)”.

-          L24-L26: “beyond TeV energies”; reference to LHC, probing centre of mass equivalent energies of 500 TeV.

-          L27: “subject to magnetic fields … and highest energies.”; maybe “subject to interactions with electromagnetic fields and cosmic matter, there possibly providing indirect information about …”; what are “highest energies”?

-          L30: “which can interact or decay”; this sounds generic and maybe misleading. They possibly interact with the atmosphere, but do decay when not disintegrated.

-          L32: “decay into gammas”; this is physics slang and should be stated more precise: “decay into gamma-ray photons with a spectrum that is …-shaped”.

-          L32: Remove “so-called”.

-          L33: “can showers induced by CRs…” -> “showers induced by CRs can …”

-          L34: What is efficiently? In what terms? The sampling here notes that not the full shower is recorded but only every 1.5 km; this should be noted in the text.

-          L34: “surface detectors”: introduce acronym here (SD), not in L47.

-          L35: At which wavelengths is the fluorescence light emitted? Clarify.

-          L36: What is “very high energies”? Set a value.

-          L39: “As the amount … primary energy.” -> “The amount … correlated with the energy …, and hence provides … primary energy, together with a measure of the statistical uncertainty of each single shower event.” Add at least one sentence about the energy calorimeter measurement uncertainties and how they depend on the assumed hadronic interactions.

-          L41-L44: “In this paper, … Observatory, … of the UHECR spectrum, …”; add commas and articles.

Section 2:

-          L46: “operating since 2004” -> “, operating since 2004,”

-          L46: “largest project”; in what terms? Clarify.

-          L47: Duplication of “It” in the beginning of two sentences. Maybe change.

-          L47: Acronym “SD” already introduced before, see Sec. 1. Use SD here.

-          L49: Remove “Therefore”.

-          L52: “1600 water-Cerenkov detectors” -> “detectors,”; later in the text, Sec. 3.6., there are 1660 detectors mentioned. Clarify which number is correct.

-          L56: “unit, the so-called vertical equivalent muon or VEM” -> “unit, which is called the vertical equivalent muon, or VEM”

-          L59: “FD” acronym already introduced before. Use FD here.

-          L60: “as it is shown” -> “as shown”

-          L63: “(Figure 1 (right))” -> “(Figure 1, right)” (several times in the text)

-          L64: “However, it can only …”; this sounds devaluating. Maybe just say: “It operates during clear moonless nights for a duty cycle of 15%.”

Section 3:

-          It appears rather arbitrary which parameters are chosen to estimate the size of events. The author should add a few sentences why exactly these few values of the LDF are chosen, and not the full information.

-          L87: “From the plot we can see some” -> “In the plot, we illustrate two”

-          L87: Maybe add an example of how many events actually were detected during the time of the measurements for Figure 2, right. E.g. the last point only is from 4 events.

-          L89: Explain how the statistical uncertainty is inferred and where the systematics come from.

-          Figure 2, left: It seems that the top x-axis labels are missing which would translate the lg(E/eV) into EeV values or similar. Related to this, it is difficult to switch between 10^X,Y values and floating EeV values (e.g. in the text: 5.08 EeV = 10^18.65 eV). Please use a consistent way to quote these numbers, I suggest both.

-          L90-L94: It is unclear what the total spectral model is, that was used to fit the data. The author explains in parts the broken power-law, but not what the suppression is. It appears like an exponential cut-off power-law, but which is not the case, according to Ref. [7], Eq. (4.1).

-          L92: “eV” -> “EeV”; Change units.

-          L94: “classical GZK scenario”: The author should first explain in one sentence what the GZK scenario is (similar to the abstract), and then quote the numbers.

-          It should be noted in 3.1., that each event, according to the procedure explained in Figure 1., has its own response function, which has to be taken into account when doing the spectral fits. Add a comment in the text.

-          L96: “The suppression” -> “However, the suppression”

-          L99: “The atmospheric depth X_max”: From the previous section the reader gets a feeling for what X_max is, but it is not told until Sec. 3.2. Introduce the meaning before, when Figure 1 is meantioned.

-          L107: “This value, being larger than that expected for a constant mass composition…” -> “This values is larger than that of a constant mass composition”; also, it is unclear what is meant with “constant mass composition”, of what? Be more specific (see also L110 “indicating that the composition becomes heavier”).

-          L116: “of the first two moments of lnA with energy”; This connection assumes the Heitler model of EASs and should be noted in the text (e.g. Matthews 2005), otherwise this appears just as a plausibility argument.

-          L117: “groups (H, He, N and Fe) obtained from” -> “groups: H, He, N, and Fe, obtained from”; Why are these groups used? Why are groups used in general? N appears very volatile compared to C or O, for example. Explain these issues.

-          L119: “hints for a contribution (25-38% depending …” -> “hints for a contribution of 25-38% (without considerations of the systematics (which I don’t know where they come from)), depending on …”; Please clarify these numbers; with systematics, the values appear consistent with zero in any case.

-          L129: “are increased leading to a faster shower development, as would happened in heavy …” -> “are increased, which would lead to a faster shower development, compared to heavy …”

-          Figure 4.: I would suggest an example at one energy of how the fits are performed to see how this works. Also, note the hadronic models, and explain the differences in a few words.

-          L130: The last sentence seems devaluating again, maybe re-phrase to be more positive? Like there is still work to be done. But the “precise” statement is misleading, as you always want to be as precise as possible.

-          L133: “energy range of the GZK suppression”: Provide a number.

-          L135: “on distance scales of 100 Mpc”: Explain why the matter distribution is not isotropic on these scales, e.g. galaxy clustering?

-          L136: Remove “in principle”, add “to first order” at the end of the sentence.

-          L138: “These use several tools like” -> “These use different tools, such as”

-          L140: References to the publications required.

-          L145-148: The sentences are misleading, as first, the anisotropy values are quoted to be “found”, but in the following sentence, it is relativized again, being consistent with isotropy. First, quote the numbers as “measured”, and then explain what they mean.

-          L148 & L152: The p-value and the level of significance to not coincide. Which test-statistic has been used to determine the significance level?

-          L151: “extragalactic origin”; while this of course is plausible, it appears that the dipole has similarities with the solar system eclipse, producing zodiacal light at about the same contours as the mean flux rate (white band in Fig. 5, right). Could the author comment on that, or is this pure coincidence?

-          L155: “is the search for photons and neutrinos in primary CRs”; this paragraph reads confusing, as neutrinos, photons, and CRs are lumped together. Of course, the author later explains that Auger can measure neutrinos as well, but the link as being “in primary CRs” sounds weird. Re-phrase the sentences.

-          L160-L161: “so-called”, “somewhat”; Use concise language.

-          L163: “about 1% of cosmogenic neutrinos is expected”; Does this imply that one of 100 events should be from neutrinos? I guess not. I believe, the sensitivity to neutrinos is much lower, otherwise you would have already seen them? Add a comment on the sensitivity.

-          L166: “young shower”: Explain. “significant”: Quantify.

-          L170: “In this case” -> “in this case,”; “tau leptons which” -> “tau leptons, which”

-          L171: “a few tens” -> “a few ten” (or tens of?)

-          L177: “electromagnetic showers, see …” -> “electromagnetic showers (see …).”

-          L178: “the selection criteria”: Explain.

-          L180: “, see Figure” -> “, as shown in Figure”

-          Figure 6.: The author should mention in the text, that predictions and upper limits from the different observatories are now on the same order of magnitude, which would mean, that soon, we can falsify models. Also, explain shortly, what the models assume (left and right).

-          L181-L186: It is not clear how between neutrinos, cosmic rays, and photons is distinguished. Explain.

-          L185: What is a top-model model; of what? Explain shortly.

-          Sec. 3.5.: Avoid too many buzz words. “The Telescope Array” is hard to understand in this context, rephrase the sentence. “The detection of the first GW…”: This is also very generic. ‘Multi-messenger’ astronomy did not have a new era with GWs, but opened another, independent, window. The collaboration with the Pierre Auger Observatory is important, and that should be noted.

-          L204: “with an energy spectrum of”: How would the high-energy neutrinos be produced, and why so long after the merger? Provide more details.

-          As a resume, you should note, when, and at what distance such a neutron star merger would be seen in neutrinos by Auger. Right now, predictions (Figure 4, right) for a NS merger at 40 Mpc are about 1 to 4 orders of magnitude below the sensitivity for neutrinos. This implies that only events in the Milky Way or the magellanic clouds could be seen. This must be clarified in the text.

-          Figure 7.: The side proportions of both plots are stretched, which make them appear in a non-professional way. Use initial proportions.

-          L212: “However, to identify …”: This is the most crucial point in running Auger, and should be noted already earlier in the text. It is key to identify the sources of UHECRs.

-          L213: “Unfortunately, the low duty cycle of the FD does not allow collecting … for energies above …” -> “The duty cycle of the FD of 15% allows collecting … only for energies below …”

-          L224: “maximum” -> “maxima”

-          L227: “for ongoing” -> “for the ongoing”

-          L231: “2019 and it will” -> “2019, and will”

-          L231: What is special about the year 2025? Explain.

Section 4:

-          L234: “change in the shape of the energy spectrum and of the” -> “change in both, the shape of the energy spectrum, and of the”

-          L236: “dipole anisotropy” -> “dipole anisotropy at E > 8 EeV”

-          L239: “The photon limits …”: The sentence has no clear structure, re-phrase.

Minor issues:

-          It appears that the Pierre Auger Observatory is not unique named in the text, sometimes “Auger”, sometimes “the Observatory”, sometimes “Auger Observatory”. I would suggest a uniform naming or abbreviation.

-          Use both, lg-energies as well as EeV values to note specific values.

-          I don’t know if the abbreviations section is required, but I would not use it, as it is not complete, and not all acronyms are used throughout.

Author Response

Response to Reviewer 2 Comments

Dear Reviewer,

Thanks very much  for valuable comments and please find below our answers.
We also attached to this response the pdf file: updated_manuscript.pdf
where in red you can find our modifications/changes in order  to address your comments.

 The text in blue will be deleted from the paper.

Best wishes

D. Gora

P.S. The line numbers  points to  the numbering scheme used
       in the file:  updated_manuscript.pdf

       Below our answer for your comments are marked by red fonts.

Point 1:  Abstract:

 -     L6: “caused by the propagation of cosmic rays”: for those in the know, it is clear that the GZK cut- off is meant, however a few more words on this already in the abstract would help to   understand the issue

  -          L7: What is generally accepted as UHECR, i.e. at what energies?

  -          L12: “super heavy particles” -> “super heavy, non-standard-model, particles”

Response 1:

     -  in the abstract we added a few more words about GZK cut-off, see L7-L9

     -  in L2 we put the energy for UHECR

     -  we replaced “super heavy particles” by  “super heavy, non-standard-model, particles” 

Point 2:  Section 1:

-          L17/L23: “detection” -> “measurement”

The text was replaced

-          L17: “… is important as it may allow us to answer …”; importance is relative to the reader, I would suggest: “… is a unique tool to provide answers to some of …”

The text was replaced

-          L21: “for at last 50 years”; reference required.

The reference (1) has been added e.g.
Linsley, J. Evidence for a Primary Cosmic-Ray Particle with Energy 1020eV.
Physical Review Letters 1963, 10,
146, doi:10.1103/PhysRevLett.10.146

-          L21: “mainly due to … in their understanding”; the sentence sounds not assuring, maybe: “because at these energies, inference is limited by statistics, (number here)”.

The text  was re-phrased and the sentence how many events we expect was added to the text see L25-26

-          L24-L26: “beyond TeV energies”; reference to LHC, probing centre of mass equivalent energies of 500

TeV.

The reference (2) has been added and text was slightly modified, see L28

-          L27: “subject to magnetic fields … and highest energies.”; maybe “subject to interactions with electromagnetic fields and cosmic matter, there possibly providing indirect information about …”; what are “highest energies”?

The text  was slightly re-phrased to avoid confiusion, L31-L32

-          L30: “which can interact or decay”; this sounds generic and maybe misleading. They possibly interact with the atmosphere, but do decay when not disintegrated.

The text was re-phrased

-          L32: “decay into gammas”; this is physics slang and should be stated more precise: “decay into gamma-ray photons with a spectrum that is …-shaped”.

The text was corrected L40

-          L32: Remove “so-called”.

Done

-          L33: “can showers induced by CRs…” -> “showers induced by CRs can …”

Done

-          L34: What is efficiently? In what terms? The sampling here notes that not the full shower is recorded but only every 1.5 km; this should be noted in the text.

Done, see L48 and also footnote 1

-          L34: “surface detectors”: introduce acronym here (SD), not in L47.

Done

-          L35: At which wavelengths is the fluorescence light emitted? Clarify.

Wavelengths for  the fluorescence light  was added in the text, see L50

-          L36: What is “very high energies”? Set a value.

Value was added,  see L51

-          L39: “As the amount … primary energy.” -> “The amount … correlated with the energy …, and hence provides … primary energy, together with a measure of the statistical uncertainty of each single shower event.” Add at least one sentence about the energy calorimeter measurement uncertainties and how they depend on the assumed hadronic interactions.

A few sentences about the energy calorimetric measurement uncertainties, has been added to the text,
 especially a few words  about the so-called  invisible energy and influence of assumed hadronc models,  
see  L52-L62

-          L41-L44: “In this paper, … Observatory, … of the UHECR spectrum, …”; add commas and articles.

Commas were added, see L71

Response 2: In general, to adress  your  comments,   the  introduction  section was re-written  
e,g.  see L34 - L41 and  L47 - L62. The red text is going to be used in the  new version of the paper,
while the text  in  blue  will be removed. Your sugestions for replacement/remove of the text has been implemented

Point 3:  Section 2:

-          L46: “operating since 2004” -> “, operating since 2004,”

The text was corrected according to your suggestion

-          L46: “largest project”; in what terms? Clarify.

This sentense was slighty modified to avoid this problem, see L76

-          L47: Duplication of “It” in the beginning of two sentences. Maybe change.

This sentense was slighty modified to avoid this problem

-          L47: Acronym “SD” already introduced before, see Sec. 1. Use SD here.

Now ‘‘SD‘‘ used in the text

-          L49: Remove “Therefore”.

Removed

-          L52: “1600 water-Cerenkov detectors” -> “detectors,”; later in the text, Sec. 3.6., there are 1660 detectors mentioned. Clarify which number is correct.

Should be 1660, and now  this number is used in the paper

-          L56: “unit, the so-called vertical equivalent muon or VEM” -> “unit, which is called the vertical equivalent muon, or VEM”

Done

-          L59: “FD” acronym already introduced before. Use FD here.

FD used in the text

-          L60: “as it is shown” -> “as shown”

Done

-          L63: “(Figure 1 (right))” -> “(Figure 1, right)” (several times in the text)

Corrected

-          L64: “However, it can only …”; this sounds devaluating. Maybe just say: “It operates during clear moonless nights for a duty cycle of 15%.”

Your has been used in the text

Response 3: In general to  adress above comments the text  was slightly  rewritten  e,g.  see L76 - L80. 
Again the text  in red is going to be used in the  new version of the paper, but  the text  in  blue  will be removed.

Your sugestions for replacemt/remove of the text has been implemented.

Point 4:  Section 3:

-          It appears rather arbitrary which parameters are chosen to estimate the size of events. The author should add a few sentences why exactly these few values of the LDF are chosen, and not the full information.

A footnote 1 has been added, in order  to explain how energy estimator for SD was choosen in the case of  the Pierre Auger Observatory

-          L87: “From the plot we can see some” -> “In the plot, we illustrate two”

Corrected, see  L127

-          L87: Maybe add an example of how many events actually were detected during the time of the measurements for Figure 2, right. E.g. the last point only is from 4 events.

The number of events has been added to the text, see L118

-          L89: Explain how the statistical uncertainty is inferred and where the systematics come from.

   A few sentences were  added to  explain this issue, see L122 - L126

-          Figure 2, left: It seems that the top x-axis labels are missing which would translate the lg(E/eV) into EeV values or similar. Related to this, it is difficult to switch between 10^X,Y values and floating EeV values (e.g. in the text: 5.08 EeV = 10^18.65 eV). Please use a consistent way to quote these numbers, I suggest both.

The Figure 2 (left) was replaced by a new one with different scale on x-axis i.e. in eV 

-          L90-L94: It is unclear what the total spectral model is, that was used to fit the data. The author explains in parts the broken power-law, but not what the suppression is. It appears like an exponential cut-off power-law, but which is not the case, according to Ref. [7], Eq. (4.1).

The text was slightly change to avoid confiusion (see L131-L132), also the footnote 2 was added in order
 to explain more precisally what kind of the fit has been performmed above the knee

-          L92: “eV” -> “EeV”; Change units.

Done

-          L94: “classical GZK scenario”: The author should first explain in one sentence what the GZK scenario is (similar to the abstract), and then quote the numbers.

The  short sescription of GZK scenarion was added to the text, see L136-L137

-          It should be noted in 3.1., that each event, according to the procedure explained in Figure 1., has its own response function, which has to be taken into account when doing the spectral fits. Add a comment in the text.

This comment was added to the text, see L121-L122

-          L96: “The suppression” -> “However, the suppression”

Done

-          L99: “The atmospheric depth X_max”: From the previous section the reader gets a feeling for what X_max is, but it is not told until Sec. 3.2. Introduce the meaning before, when Figure 1 is meantioned.

The defintion of X_max is now introduced, see L142

-          L107: “This value, being larger than that expected for a constant mass composition…” -> “This values is larger than that of a constant mass composition”; also, it is unclear what is meant with “constant mass composition”, of what? Be more specific (see also L110 “indicating that the composition becomes heavier”).

The text was slightly modified

-          L116: “of the first two moments of lnA with energy”; This connection assumes the Heitler model of EASs and should be noted in the text (e.g. Matthews 2005), otherwise this appears just as a plausibility argument.

The footnote 3 and reference to (e.g. Matthews 2005),   was added to the text, see L160

-          L117: “groups (H, He, N and Fe) obtained from” -> “groups: H, He, N, and Fe, obtained from”; Why are these groups used? Why are groups used in general? N appears very volatile compared to C or O, for example. Explain these issues.

The text was re-written to avoid confusion, see  L161-L171. Why  groups: H, He, N, Fe because this is result of the fitting to the Auger X_max  data.

-          L119: “hints for a contribution (25-38% depending …” -> “hints for a contribution of 25-38% (without considerations of the systematics (which I don’t know where they come from)), depending on …”; Please clarify these numbers; with systematics, the values appear consistent with zero in any case.

The text was re-written to avoid confusion,  and this statment was removed, see  again L161-L171.

-          L129: “are increased leading to a faster shower development, as would happened in heavy …” -> “are increased, which would lead to a faster shower development, compared to heavy …”

Corrected in L177-L178

-          Figure 4.: I would suggest an example at one energy of how the fits are performed to see how this works. Also, note the hadronic models, and explain the differences in a few words.

Sorry I would like to keep this figure because it shows  important results  from the Auger Obserwatory,

and  it was also presented in the ICNP 2018 conference. A few words about intercation models were added in the text, see again   L161-L171

-          L130: The last sentence seems devaluating again, maybe re-phrase to be more positive? Like there is still work to be done. But the “precise” statement is misleading, as you always want to be as precise as possible.

The sentence was re-phrased, see L178-L179

-          L133: “energy range of the GZK suppression”: Provide a number.

The number was added, see L182

-          L135: “on distance scales of 100 Mpc”: Explain why the matter distribution is not isotropic on these scales, e.g. galaxy clustering?

Yes due to galaxy clustering, the sentence was slightly modified, see L182-L183

-          L136: Remove “in principle”, add “to first order” at the end of the sentence.

Done in L186

-          L138: “These use several tools like” -> “These use different tools, such as”

Done in L187

-          L140: References to the publications required.

Done, references were added, see L189

-          L145-148: The sentences are misleading, as first, the anisotropy values are quoted to be “found”, but in the following sentence, it is relativized again, being consistent with isotropy. First, quote the numbers as “measured”, and then explain what they mean.

These  sentences were  re-phrased,  see L194 – L199

-          L148 & L152: The p-value and the level of significance to not coincide. Which test-statistic has been used to determine the significance level?

The p-value= 2.6×10−8 is equivalent to a two-sided Gaussian significance of 5.6σ

 Allowing for a penalization factor of 2 to account for the fact that two energy bins were explored, the significance is reduced  to 5.4σ.  Further penalization for the four additional lower energy bins examined in [Astropart.Phys.34:627-639,2011]  which falls to 5.2σ.

The definition of test-statisctic and the full method can be found in [Astropart.Phys.34:627-639,2011],
but this da  ratio of the likelihood for  the signal and the background case.

We include this explanation in the text, see again see L195 - L200

-          L151: “extragalactic origin”; while this of course is plausible, it appears that the dipole has similarities with the solar system eclipse, producing zodiacal light at about the same contours as the mean flux rate (white band in Fig. 5, right). Could the author comment on that, or is this pure coincidence?

For me just coincidence

-          L155: “is the search for photons and neutrinos in primary CRs”; this paragraph reads confusing, as neutrinos, photons, and CRs are lumped together. Of course, the author later explains that Auger can measure neutrinos as well, but the link as being “in primary CRs” sounds weird. Re-phrase the sentences.

The sentence was re-phrased, “in primary CRs”  was removed, see L206

-          L160-L161: “so-called”, “somewhat”; Use concise language.

Removed from the text

-          L163: “about 1% of cosmogenic neutrinos is expected”; Does this imply that one of 100 events should be from neutrinos? I guess not. I believe, the sensitivity to neutrinos is much lower, otherwise you would have already seen them? Add a comment on the sensitivity.

The comment about senstivity was added  in L214-L215

-          L166: “young shower”: Explain. “significant”: Quantify.

The sentece was slightly modified , see L218-220

-          L170: “In this case” -> “in this case,”; “tau leptons which” -> “tau leptons, which”

Done

-          L171: “a few tens” -> “a few ten” (or tens of?)

Done

-          L177: “electromagnetic showers, see …” -> “electromagnetic showers (see …).”

Done

-          L178: “the selection criteria”: Explain.

Done

-          L180: “, see Figure” -> “, as shown in Figure”

Done

-          Figure 6.: The author should mention in the text, that predictions and upper limits from the different observatories are now on the same order of magnitude, which would mean, that soon, we can falsify models. Also, explain shortly, what the models assume (left and right).

A few sentences were added about this, see L232-L237

-          L181-L186: It is not clear how between neutrinos, cosmic rays, and photons is distinguished. Explain.

A few sentences were added about this, see L238-L241

-          L185: What is a top-model model; of what? Explain shortly.

A short  explanation  was added, see  L246-L250

-          Sec. 3.5.: Avoid too many buzz words. “The Telescope Array” is hard to understand in this context, rephrase the sentence. “The detection of the first GW…”: This is also very generic. ‘Multi-messenger’ astronomy did not have a new era with GWs, but opened another, independent, window. The collaboration with the Pierre Auger Observatory is important, and that should be noted.

Some text was deleted , ad we replace word ‚‘‘era‘‘ by  ‘‘window‘‘, see L254

-          L204: “with an energy spectrum of”: How would the high-energy neutrinos be produced, and why so long after the merger? Provide more details.

A few sentences were added about this, see L261-L265

-          As a resume, you should note, when, and at what distance such a neutron star merger would be seen in neutrinos by Auger. Right now, predictions (Figure 4, right) for a NS merger at 40 Mpc are about 1 to 4 orders of magnitude below the sensitivity for neutrinos. This implies that only events in the Milky Way or the magellanic clouds could be seen. This must be clarified in the text.

This comment was added to the text, see L275-L276

-          Figure 7.: The side proportions of both plots are stretched, which make them appear in a non-professional way. Use initial proportions.

Corrected

-          L212: “However, to identify …”: This is the most crucial point in running Auger, and should be noted already earlier in the text. It is key to identify the sources of UHECRs.

-          L213: “Unfortunately, the low duty cycle of the FD does not allow collecting … for energies above …” -> “The duty cycle of the FD of 15% allows collecting … only for energies below …”

The text was corrected according to above sugestions

-          L224: “maximum” -> “maxima”

Corrected

-          L227: “for ongoing” -> “for the ongoing”

Corrected

-          L231: “2019 and it will” -> “2019, and will”

Corrected

-          L231: What is special about the year 2025? Explain.

The sentence was  slightly modfied,  the  text:
‘‘and it will be followed by data-taking  until  2025.‘‘ was removed.

Response 4:

In general to  adress above comments the text  was slightly  re-written  
Again the text  in red is going to be used in the  new version of paper, but  the text  in  blue  will be removed.

Your sugestions for replacemt/remove of the text has been implemented.

Point 5:  Section 4:

-          L234: “change in the shape of the energy spectrum and of the” -> “change in both, the shape of the energy spectrum, and of the”

Done

-          L236: “dipole anisotropy” -> “dipole anisotropy at E > 8 EeV”

Done

-          L239: “The photon limits …”: The sentence has no clear structure, re-phrase.

 The sentence was re-phrase

Point 6:  Minor issues:

-          It appears that the Pierre Auger Observatory is not unique named in the text, sometimes “Auger”, sometimes “the Observatory”, sometimes “Auger Observatory”. I would suggest a uniform naming or abbreviation.

Done

-          Use both, lg-energies as well as EeV values to note specific values.

I will keep like this.

-          I don’t know if the abbreviations section is required, but I would not use it, as it is not complete, and not all acronyms are used throughout.

The abbrevation section was removed
